# Detection and Analysis of Pavement-Section Based on Laser Displacement Sensor

**DOI:** 10.3390/s23156758

**Published:** 2023-07-28

**Authors:** Jin Han, Gao Xiong, Jia Liu

**Affiliations:** School of Mechanical Engineering, Shaanxi University of Technology, Hanzhong 723001, China; xionggao0505@163.com (G.X.); isabell1016@163.com (J.L.)

**Keywords:** laser displacement sensor, vehicle-mounted high-speed detection, sensor fusion, section detection, filter processing

## Abstract

The section detection of the pavement is the data basis for measuring the road smoothness, rutting, lateral slope, and structural depth. The detection of the Pavement-Section includes longitudinal-section inspection and cross-section inspection. In this paper, based on multiple laser displacement sensors, fused accelerometers and attitude sensors, and using vehicle-mounted high-speed detection, we design a sensor-fused pavement section data acquisition method, establish the relevant mathematical model, and realize the automatic acquisition of pavement longitudinal and transverse sections. The acceleration sensor is filtered to improve the accuracy of data acquisition, and the error of the detection system is calculated and analyzed. Through the actual measurement, the vehicle-mounted high-speed pavement profile detection method adopted in this paper can not only accurately detect the profile of the pavement profile, but also improve the detection efficiency, providing a cost-effective detection mode for road surface detection.

## 1. Introduction

With the continuous development of transportation, as of 2022, the total road mileage in the world has reached 65 million kilometers [1]. Therefore, the quality of the road surface is particularly important for the smooth operation of the road and the safe operation of vehicles [2]. Pavement section testing is the main basis for pavement quality testing [3]. Regular inspection of the road section can identify problems such as pavement level, rutting, lateral slope and structural depth in time for timely repair and maintenance to ensure the levelness and durability of the road [4]. In addition, through the detection of Pavement-Sections, the load-carrying capacity can also be evaluated to provide a basis for reasonable road maintenance and management decisions [5]. The detection of Pavement-Sections includes longitudinal-section detection and cross-section detection [6]. Pavement profile detection is the detection of the vertical plane cut along the longitudinal direction of the road [7], which mainly reflects the road surface undulation of the road in the medium and small wavelengths, and is used to detect the smoothness of the road surface [8]; Pavement cross-section detection is a plane detection perpendicular to the longitudinal-section of the road [9], which mainly reflects the lateral undulation of the road. It is used to detect the rutting and structural depth of the road surface [10] and can also detect the lateral slope of the road surface [11]. The commonly used cross-section measurement methods mainly include the level tape measure method [12], theodolite line-of-sight method [13], current total station measurement method [14], GPS measurement [15], etc.

These methods are effective in practice, but they also have some defects and limitations. Here are some common defects:(1)The detection accuracy is low: the level tape measure method, theodolite line-of-sight method, total station measurement method, and GPS measurement are all single-device measurements [16], which will affect the data collection accuracy due to equipment errors and environmental disturbances (rain, fog and the environment) [17]. The level tape measure method and the theodolite line-of-sight method are difficult to guarantee accuracy due to manual reading [18].(2)Limited coverage: although GPS measurement can provide wide-area data, it cannot be detected in signal-blind areas such as mountain roads and tunnels [19], and the resolution is not enough to obtain detailed Pavement-Section information [20].(3)Time and labor costs: the level tape measure method and theodolite sight distance method require manual measurement of road closures during measurement, and the time and labor costs are relatively high [21].(4)Devices such as laser cross-sections based on laser displacement sensors can only detect the cross-section of the pavement and have large data errors due to the lack of data fusion from sensors such as accelerometers and odometers.(5)Existing vehicle-mounted laser detection equipment can only detect one road width in real-time, which has certain limitations.

In response to the shortcomings of today’s pavement section detection [22], this paper designs a pavement section detection system that integrates multiple sensors, combines the advantages of various sensors, and designs a pavement section data acquisition method based on the fusion of multiple laser displacement sensors using an on-board high-speed non-contact automatic detection method. This method can effectively solve various problems of today’s road inspection, such as signal loss and accuracy problems of GPS, error problems of single sensors, safety and efficiency problems of manual measurement, etc. It provides a new model for the intelligence of current pavement inspection.

## 2. Main Research Content

### 2.1. The Principle of Pavement Section Profile Data Acquisition

Aiming at the detection requirements, this paper remodels a medium-sized bus and installs a detection steel beam at the rear of the bus. The detection beam is made of stainless steel, which can prevent corrosion and work in poor weather. It is mainly used to install multi-channel laser displacement sensors and accelerometers. Complete the acquisition of pavement longitudinal and cross-sectional profile data. The detection beam is horizontally installed at the bottom of the rear end of the detection vehicle, and the actual detection beam is shown in Figure 1. Multi-channel laser displacement sensor road profile acquisition is shown in Figure 2.

In addition, the road surface detection vehicle is also equipped with a photoelectric encoder to measure the vehicle speed *v*(*t*) and distance *s*(*t*); the attitude sensor accurately collects the detection vehicle attitude parameters (roll angle *ϕ*, pitch angle *θ*, and heading). The temperature and humidity sensor collects the temperature and humidity of the environment [23], and the detection principle is shown in Figure 3.

### 2.2. Pavement Longitudinal-Section Data Acquisition

When collecting the longitudinal-section data of the road surface, the laser displacement sensor installed on the detection beam at the rear of the detection vehicle and three sets of acceleration sensors (the wheel lines of the left and right wheels and the center line of the detection vehicle) are used to measure and collect the road longitudinal profile. The sampling frequency setting of the accelerometer is set at 0.5 Hz–2 kHz according to the driving speed of the inspection vehicle to ensure the accuracy of sampling.

As shown in Figure 4, the longitudinal profile detection is the longitudinal undulation value Δh(t). First, a measurement datum is calibrated; then, the relative distance $h_{4}(t)$ is obtained by scanning the road surface at high-speed using multiple laser displacement sensors, and the longitudinal displacement $h_{2}(t)$ is obtained by accelerometers; finally, Δh(t) is obtained by geometric calculation of the elevation and pitch angle $\theta_{x}(t)$ of the vehicle detection beam.
(1)$$\Delta h(t)=h_{1}-h_{0}-h_{3}(t)=h_{1}-h_{0}-[h_{4}(t)\cdot \cos(\theta_{x}(t))-h_{2}(t)]$$

In Equation (1):

*t*: the detection interval time, the value of t is related to the sampling frequency of the accelerometer, and the unit is ms;

$\theta_{x}(t)$: the pitch angle of the detection beam is measured by the attitude sensor on the vehicle, and its value is related to the structure of the vehicle body, road conditions, and the driving state of the vehicle, and the unit is °;

*h*_0_: calibrate the initial value of the detection beam in the coordinate system before measurement, which is a constant and the unit is mm;

*h*_1_*:* the longitudinal displacement at the previous moment, in mm;

*h*_2_(*t*): the longitudinal displacement obtained from the acceleration sensor on the detection beam relative to the previous moment, and the unit is mm;

*h*_4_(*t*): The distance between the detection beam and the road surface, the unit is mm.

Therefore, to calculate Δh(t), it is necessary to calculate the value of *h*_2_(*t*) at any time. According to the relationship between speed, displacement, and acceleration, it can be obtained:(2)$$\begin{array}{ll} h_{2}(t) & =h_{2}(0)+\int_{0}^{t}v(t)dt=h_{2}(0)+v(0)t+\int_{0}^{t}\int_{0}^{t}h_{2}\prime \prime (t)dtdt\\ & =h_{2}(0)+v(0)t+\int_{0}^{t}\int_{0}^{t}a(t)\cos(\theta_{x}(t))dtdt \end{array}$$

In Equation (2), *h*_2_(0) is a constant, which does not affect the profile curve shape of the road surface, *a*(*t*) is the acquired value of the accelerometer, and *v*(0) is a slope linear function superimposed on *h*_2_(*t*), which is equivalent to a sloped road surface and needs to be calculated. The least squares method [24] can be used to calculate *v*(0), that is, it is considered that the overall trend of the measured road surface between the measurement start point and the endpoint is a flat road with a slope of 0. The calculation process of *v*(0) is shown in Figure 5.

Since the attitude sensor can measure the value of $\theta_{x}(t)$, the value of *v*(0) can be calculated according to Figure 5, so that the formula $\Delta h\left(t\right)$ of the profile of the road surface can be deduced. See Equation (3).
(3)$$\Delta h(t)=h_{1}-h_{0}+h_{2}(0)+v(0)t-h_{4}(t)\cdot \cos(\theta_{x}(t))+\int_{0}^{t}\int_{0}^{t}a(t)\cos(\theta_{x}(t))dtdt$$

### 2.3. Pavement Cross-Section Data Acquisition

The laser displacement sensor configured on the road surface inspection vehicle can not only obtain the data of the longitudinal section of the road surface but also use the detection beam for lateral correlation to obtain the cross-section data of the road surface. The working principle is shown in Figure 6.

It can be seen from the schematic diagram 6 that the contour curve of the road cross-section can be constructed by solving the relative intercept Δ*y* of the road cross-section, which can be obtained from Figure 6:(4)$$\Delta y=y_{2}-y_{1}=x_{0}\sin\left(\varphi_{x}\right)+y_{3}\cos\left(\varphi_{x}\right)-y_{4}\cos\left(\varphi_{x}\right)$$

In Equation (4):

*x*_0_: the distance between any two laser displacement sensors of the detection beam is a known quantity, the unit is mm;

$x_{0}\sin\left(\varphi_{x}\right)$: coordinate system distance between two laser displacement sensors, the unit is mm;

$\varphi_{x}$: the detected vehicle roll angle obtained by the attitude sensor, the unit is °;

*y*_3_, *y*_4_: the laser displacement sensor measures the distance between the road surface and the detection beam, the unit is mm.

## 3. Error Processing of Data Collected by Sensors

### 3.1. Sources of Error in Acceleration Signals

Acquisition of acceleration signals has used the method of temperature and humidity compensation to reduce the influence of the environment on the signal, but when the acceleration is used to calculate the velocity and displacement, there are also interferences from DC-components, integral accumulation, and trend items, so it is necessary to calculate the velocity and displacement of the acceleration time algorithm optimization. The specific optimization algorithm flow chart is shown in Figure 7.

It can be seen from the flow chart that the main ways of errors in the process of acceleration restoration and integration are as follows:(1)DC-component: Vehicle-mounted accelerometers are measured by electrical signals when collecting data, which is prone to errors caused by DC-components introduced by line interference, digital-to-analog conversion, and filtering.(2)Integral accumulation: Acceleration obtains velocity and displacement through integration, and different integration algorithms will produce different algorithm errors.(3)Trending item: Multiple integration and measurement vibrations will generate trend items, which will have a greater impact on the accuracy of the integration results.

### 3.2. Error Process of Acceleration Signal

(1)Remove DC-component

The mean value of the collected acceleration signal is used as an estimate of the DC-component, which is separated from the collected original signal [25]. The specific method is to first calculate the average value $\bar{a}$ of the collected acceleration, and then subtract the average value from each acceleration value to obtain the acceleration $a_{i}^{\prime }$ after removing the DC-component. This method can also be used to remove the DC-component of the velocity.
(5)$$a_{i}^{\prime }=a_{i}-\bar{a}\;\left(i=0,\ldots ,n-1\right)$$

In Equation (5)
$$\bar{a}=\frac{1}{n}\sum_{i=0}^{n-1}a_{i}$$

(2)Choice of Digital Integral

The acceleration signal is digitally integrated once to obtain the velocity and twice integrated to obtain the displacement. The acceleration signal can be digitally integrated after the DC-component is removed, and the trapezoidal integration method, Simpson integration method [26], etc. can be used. By comparison, the calculation accuracy of the Simpson integral method is high, and the program is easy to implement, so this paper uses Simpson integral method to realize the calculation of velocity and displacement.

The mathematical expression of the Simpson integral is:(6)$$y\left(k\right)=\frac{\Delta t}{6}\sum_{i=1}^{k}x\left(i-1\right)+4x\left(i\right)+x\left(i+1\right)\;\;\;\;\;\;\;k=1,2,...,N-1$$

Thus, the formula for calculating the velocity using the Simpson method can be obtained as follows:(7)$$v\left(k\right)=v\left(k-1\right)+\frac{\Delta t}{6}\left[a\left(k-1\right)+4a\left(k\right)+a\left(k+1\right)\right]\;\;\;\;\;\;\;\;k=1,2,...,N-1$$

The formula for calculating displacement is as follows:(8)$$s\left(k\right)=s\left(k-1\right)+\frac{\Delta t}{6}\left[v\left(k-1\right)+4v\left(k\right)+v\left(k+1\right)\right]\;\;\;\;\;\;\;k=2,...,N-2$$

Construct a simple standard function to verify the accuracy of Simpson’s algorithm, setting the acceleration as a sine function:(9)$$a=2\sin\left(10\pi t\right)$$

When the sampling frequency is 22 kHz, the original acceleration signal curve and the displacement curve obtained after the acceleration signal is integrated twice according to the Simpson method are shown in Figure 8 and Figure 9.

From Figure 9, it can be obtained that the magnitude of the integrated displacement is max = 0.00202855 mm, min = −0.00202852 mm.
(10)$$\frac{2}{{\left(10\pi \right)}^{2}}=0.00202848\;mm$$

The algorithmic error of displacement calculated by Simpson’s numerical integration method is about 0.003%. According to the actual calculation, the maximum algorithmic error produced by the second integration is 0.32 mm.

(3)Remove trending items

Since the accelerometer acquires data in the form of a vehicle, vibrations, bumps, and magnetic field interference from vehicle electrical appliances during vehicle driving it will cause the integrated speed and displacement signals to deviate from the baseline [27], and the amplitude of the deviation from the baseline will change at any time with the increase in the detection distance. This phenomenon is a trend term. When the data is integrated, the trend item will have a great impact on the integration result, and the obtained velocity and displacement results may be completely distorted. Therefore, eliminating the trend item is an important part of acceleration to restore the velocity and displacement signals. The polynomial least squares method is a method with higher precision for eliminating trend items, and its steps are as follows:

**Step 1:** A polynomial function is constructed from the aggregated values of the acceleration signal $a\left(1,2,\ldots ,i\right)$ and its corresponding time series $t\left(1,2,\ldots ,i\right)$.
(11)$$\tilde{a}\left(k\right)=b_{0}+b_{1}t\left(k\right)+b_{2}t^{2}\left(k\right)+...+b_{n}t^{n}\left(k\right)\;\;\;\;\;\;\;\;\left(k=1,2,...,i\right)$$

**Step 2:** Determine each undetermined coefficient $b_{0},b_{1},b_{2},\ldots ,b_{m}$ of the function $\tilde{a}(k)$, so that the sum of the squares of the error between the discrete data *a* and the function $\tilde{a}(k)$ is the smallest, that is:(12)$$E={\sum_{k=1}^{i}\left[a-\tilde{a}\left(k\right)\right]}^{2}$$

According to the principle of extremum, solve $b_{0},b_{1},b_{2},\ldots ,b_{n}$ that satisfies the minimum value of *E*.

**Step 3:** Let, $\frac{\partial E}{\partial b_{i}}=0\;\left(i=0,1,...,n\right)$ get *n* + 1 element linear equations, solve the equations, can solve the value of n + 1 undetermined coefficients. The value of n is determined by the collected signal during actual detection, and an appropriate value of n is selected to process the polynomial trend item elimination for the sampled data.

Set the acceleration signal of a trend item (*n* = 1):(13)$$a=0.5\;\sin\left(100\pi t\right)+2t$$

The comparison of velocity and displacement after detrending is shown in Figure 10:

Removing the velocity trend term is most effective when the value of n is set to 2; when n is set to 3, removal of the displacement trend term is most effective. According to Figure 10, it can be seen that by removing the trend items of acceleration and velocity, relatively correct velocity and displacement curves can be obtained. If the trend items are not removed, there will be a large overall drift of the signal, and the real information cannot be restored.

### 3.3. Error Analysis

The measurement error of the system is mainly generated by the sensor, such as the error generated by the laser displacement sensor and the accelerometer sensor, and the error of the attitude sensor. In addition, the comprehensive algorithm error also affects the error of the whole system.

#### 3.3.1. Sensor Measurement Error Analysis

The measurement accuracy of the acceleration sensor used in the system is $0.0001\;g\approx 0.00098\;{m/s}^{2}$ [28]. Therefore, after the second integration, the displacement error introduced by the measurement error is:(14)$$\frac{0.98}{{\left(2\pi f\right)}^{2}}=\frac{0.0248}{f^{2}}\;mm$$

In Equation (14)

*f* is the frequency of the collected signal, and the accelerometer collection range in this paper is *f* > 0.1 Hz. Therefore, the measurement error curve of the acceleration sensor can be obtained as shown in Figure 11.

From Figure 11a, it can be found that the lower the road shape acquisition frequency, the greater the error generated by the displacement after the acceleration signal integration. The maximum value of the error appears at 0.1 Hz. By calculation, the displacement error of the acceleration sensor at 0.1 Hz is:(15)$$\delta_{a}=\frac{0.0248}{f_{a}^{2}}=\frac{0.0248}{0.1^{2}}=2.48\;mm$$

The 2.48 mm here is the maximum error value measured by the acceleration sensor. Generally, the error generated by the acceleration sensor is related to the measured road surface frequency *f_r_*. There is the following relationship between the frequency *f_r_* (Hz) of the tested road surface, the driving speed of the detection vehicle (m/s), and the wavelength *λ* (m) of the road surface:(16)$$f_{r}=\frac{v}{\lambda }$$

In actual testing, the unit of the testing vehicle’s driving speed *v*’ is km/h, and after unit conversion, Equation (12) can be changed to:(17)$$f=\frac{1000v^{\prime }}{3600\lambda }=\frac{5v}{18\lambda }$$

After bringing it into Equation (11), the estimation method of the measurement error of the acceleration sensor can be obtained.
(18)$$\delta_{a}=\frac{0.0248\times 324\lambda^{2}}{25v^{2}}mm$$

Under normal circumstances, the actual measurement value of the wavelength of various types of roads is 1–100 m, and the driving speed of the detection vehicle is generally 35–80 km/h. The maximum measurement error of the acceleration sensor can be calculated to be 2.62 mm.

#### 3.3.2. Measurement Error Analysis of Attitude Sensor

The error of the measurement angle after complementary filtering according to the attitude of the detection vehicle is ±0.2 degrees, so the detection error of the Pavement-Section caused by the measurement error of the detection vehicle pitch angle and roll angle is:

Longitudinal-section error: $400\times \left[1-\cos\left(0.2^{0}\right)\right]=0.0024\;\mathrm{mm}$

Cross-section error: $1500\times \sin\left(0.2^{0}\right)+400\times \left(1-\cos\left(0.2^{0}\right)\right)=5.24\;\mathrm{mm}$

The measuring range of the laser displacement sensor is 200–400 mm, and the maximum distance between the laser displacement sensors installed on the detection beam is 1500 mm.

### 3.4. Comprehensive Error Analysis

For the centimeter-meter level road shape undulation on the road surface, when the laser displacement sensor and the acceleration sensor are used to detect the road shape, the comprehensive error is mainly composed of the sensor acquisition error and the algorithm error. In this case, the comprehensive measurement error is composed of acceleration integration algorithm error, laser displacement sensor measurement error, acceleration sensor measurement error, and measurement error caused by vehicle attitude (measurement error caused by pitch angle, measurement error caused by roll angle).

Through the error analysis and calculation of the acceleration sensor, attitude sensor and integral algorithm, it can be seen that the maximum error caused by the acceleration integral algorithm itself is 0.32 mm, and the maximum measurement error caused by the vehicle attitude is 5.25 mm (measurement error caused by pitch angle is 0.0024 mm + measurement error caused by roll angle is 5.24 mm), and the system uses a laser displacement sensor to measure a maximum error of 0.50 mm [29] (provided by the device parameters). Therefore, the comprehensive error of the system is:(19)$$\delta_{s}=\delta a+5.25+0.5+0.32=\frac{0.0248\times 324\lambda^{2}}{25v^{2}}+6.07\;mm$$

To have a more intuitive understanding of the error, formula (15) can be used to calculate the system measurement error under different vehicle speeds and different road surface waveforms, the error data is shown in Table 1. At the same time, considering that the measurement range of the laser displacement sensor is 200–400 mm, that is, when the laser and acceleration are used to measure the cross-sectional road shape, the undulation of the road does not exceed 400 mm. Then 400 mm can be used as the maximum height range of the system so that the detection accuracy of the system can be obtained as:(20)$$Detection\;precision=\frac{\delta_{s}\;(mm)}{400\;(mm)}\%$$

When the inspection vehicle measures the 20-m-wavelength road undulation at a speed of 35 km per hour, the accuracy of the full scale is about 1.54%. This accuracy will also change slightly with the increase in the vehicle speed and the change of the wavelength of the road surface. See Table 1 for details.

It can be seen from the data in the table that when detecting Pavement-Sections, the maximum measurement error is about 8.69 mm, and the maximum measurement accuracy is 2.17%, which is lower than the national detection standard (error of 10%) [5]. The average measurement accuracy is about 1.65%, which is relatively high, especially when the detection wavelength of the road surface is limited to a short range, the measurement accuracy can reach about 1.5%. Therefore, this method is suitable for the detection of relatively flat roads with small slopes and small undulations (undulations <= 400 mm) such as airport runways and expressways.

## 4. Experimental Validation

In order to validate the pavement section detection (combination of laser and acceleration), a shape comparison-based scheme is used in this paper for validation.

The method of comparing with known standard shapes is used for verification. By setting blocks of known shapes on the road, the inspection vehicle uses laser displacement sensors and acceleration sensors to measure and calculate to obtain the section curve information of the road and then analyzes the measurement accuracy and error of the system by directly comparing the similarity and error of the curves.

The experiment was designed with 2 kinds of blocks, large blocks were used to simulate the undulation of potholes on the road, using a wooden board with a length of 200 cm, width of 30 cm and height of 3 cm; small blocks were used to simulate the undulation of particles on the road, using a length of 200 cm and a semicircular wooden strip with a diameter of 1–6 cm as shown in Figure 12.

### 4.1. Verification Experiments with Large Objects

Six wooden boards were distributed at uneven intervals of 2 m, 2 m, 2 m, 3 m and 3 m. The inspection vehicle was driven at different speeds to press over the planks, and the waveform curves of the planks were finally obtained by calculating the measured data from the laser displacement sensor and acceleration sensor.

In the school to choose a straight road surface, after a number of different speed passing tests, using 30 km/h speed passing, the calculation of the road surface waveform curve is shown in Figure 13.

The results of the test at 60 km/h are shown in Figure 14.

In the test of six planks, the results of two tests with different vehicle speeds were calculated, the results of the 30 km/h test are shown in Table 2 and the results of the 60 km/h experiment are shown in Table 3.

As can be seen from the results of the above two test data comparisons, the average error in calculating the high longitudinal degree of the plank and the real plank height (30 mm) is 2.25 mm (30 km/h), 3.5 mm (60 km/h), and the maximum error is 4.8 mm (30 km/h), 6.2 mm (60 km/h); the average error in calculating the length of the plank transverse direction (3000 mm) is 2.8 mm (30 km/h), 3.5 mm (60 km/h) and the maximum error is 5.3 mm (30 km/h), 6.6 mm (60 km/h). The test results are all less than the theoretical error of 8.69 mm in Table 1, meeting the test standard.

### 4.2. Verification Experiments with Small Objects

Measurement tests were carried out using strips of wood with semicircular cross-sections, length of 2 m, diameters ranging from 1 cm to 6 cm, with an interval of 1 m between the 5 strips of wood of 1 cm diameter, and 4 strips of wood of other diameters, each with an interval of 2 m, with an interval of 5 m between the last strip of wood of 1 cm diameter and the first strip of wood of 2 cm diameter, and an interval of 4 m between the last strip of wood of different diameters and the next strip of wood of 1 cm diameter. The test vehicle was passed at many different speeds and the curve of the wooden strips obtained at a speed of 30 km/h is shown in Figure 15.

The results of one test were randomly selected from the test results of multiple wood strips. The test results of different diameters of wooden strips in the test curve were compared. The test results are shown in Table 4.

As can be seen from Table 4, the minimum error of transverse length detection is +0.8 mm at 1 cm semicircular wooden strips, and the maximum error is +7.8 mm at 6 cm semicircular wooden strips; the minimum error of longitudinal height detection is +0.8 mm at 1 cm semicircular wooden strips, and the maximum error is +6.9 mm at 6 cm semicircular wooden strips. The surface of the wooden strips used in the test is not exactly a smooth rounded curve, there is a certain height error, and the error range is about 0.5 mm. In addition, in the test, the wooden strips are placed on the ground, not completely flat with the ground, so the actual height above the ground is greater than the radius of the wooden strips (which is also an important reason for all the measurements are large), so there will be a systematic error in the test, about 2 mm. Under this systematic error factor, the measurement of the shape of the small wooden strip, the measurement error is basically within the range of 6 mm, which is less than the maximum theoretical error value of 8.69 mm in Table 1.

### 4.3. Analyze

From the experiments, it can be found that during the detection experiments based on shape error comparison. The error of large object detection is related to the speed of the detecting vehicle, and the larger the speed, the error of detection increases; when small object detection, the larger the object cross-section, the larger the error, but the detection of small objects is less sensitive to the vehicle speed. Therefore, the detection method in this paper is highly accurate at low speeds (30 km/h) for ordinary roads with large undulations in the road surface; higher speeds (≥60 km/h) can be used for detecting runways and highways with small, straight road surface particles.

## 5. Conclusions

This paper studies and analyzes the acquisition and calculation methods of pavement section data based on the fusion of laser displacement sensor, accelerometer and attitude sensor data. A vehicle-mounted high-speed pavement section detection mode is proposed, whose main contributions are as follows:(1)The longitudinal and transverse sections of the pavement can be detected in real-time, and the vehicle-mounted detection beam is equipped with multiple laser displacement sensors and can be adjusted for expansion and contraction according to the width of the pavement so that two lanes can be detected simultaneously.(2)The proposed data fusion calculation method and the filtering method of the accelerometer can improve the detection accuracy.(3)The maximum detection speed is 80 km/h and the detection efficiency is high due to the vehicle-based real-time detection.

In summary, the pavement section detection method based on laser displacement sensor proposed in this paper can effectively improve the detection accuracy and detection efficiency and is applicable to the flatness and slope detection of various types of pavements. It can meet the inspection requirements of rural roads, national roads, airport runways and highways.

## Figures and Tables

**Figure 1 sensors-23-06758-f001:**
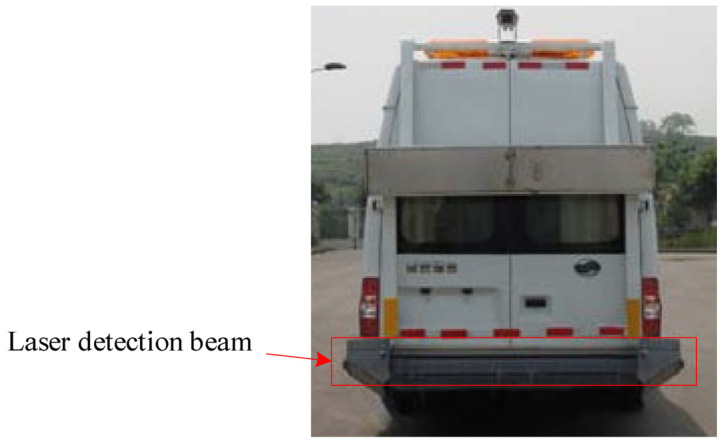
Vehicle-mounted laser detection beam.

**Figure 2 sensors-23-06758-f002:**
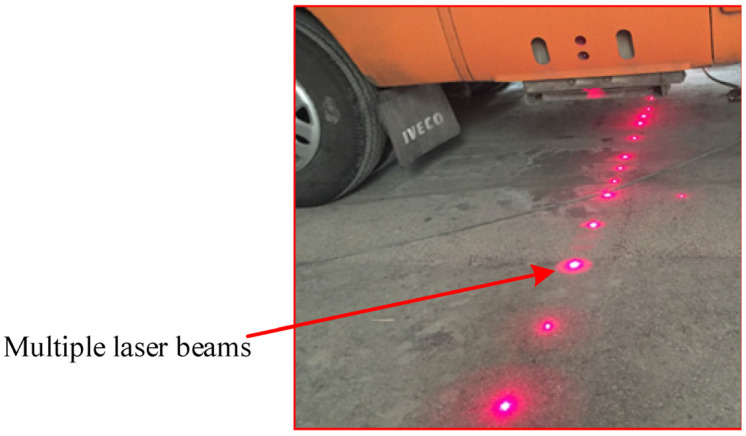
Multiple Laser Beam Acquisition.

**Figure 3 sensors-23-06758-f003:**
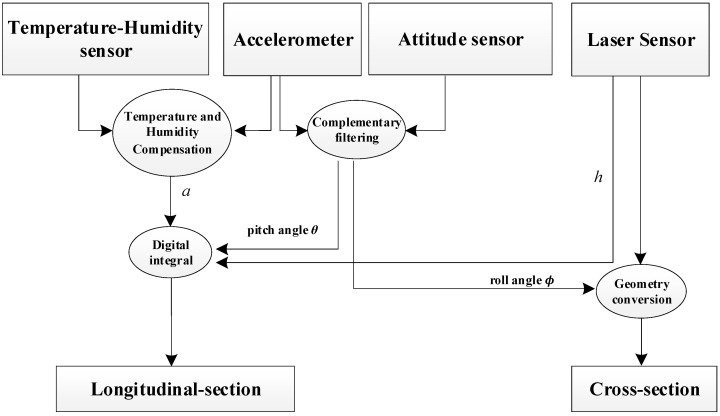
The Principle of Pavement-Section Acquisition.

**Figure 4 sensors-23-06758-f004:**
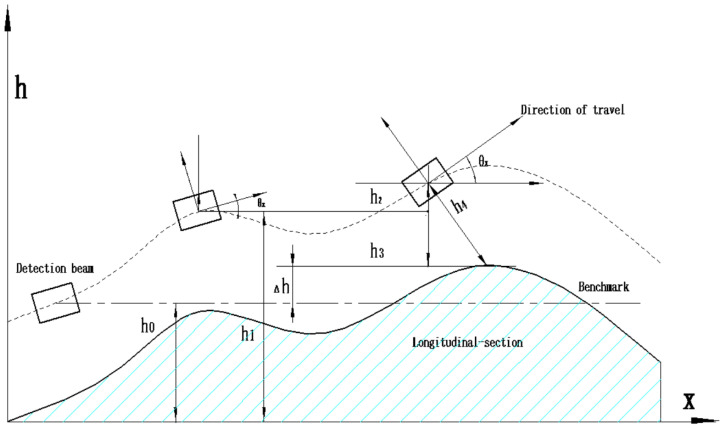
The pavement longitudinal-section Acquisition Principle.

**Figure 5 sensors-23-06758-f005:**
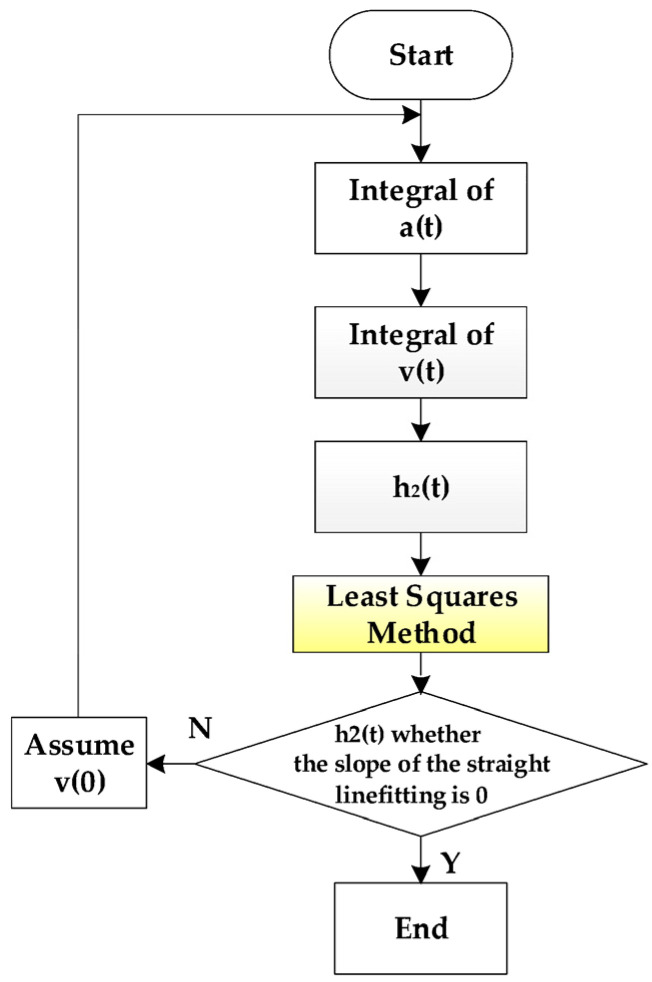
The *v*(0) calculation process.

**Figure 6 sensors-23-06758-f006:**
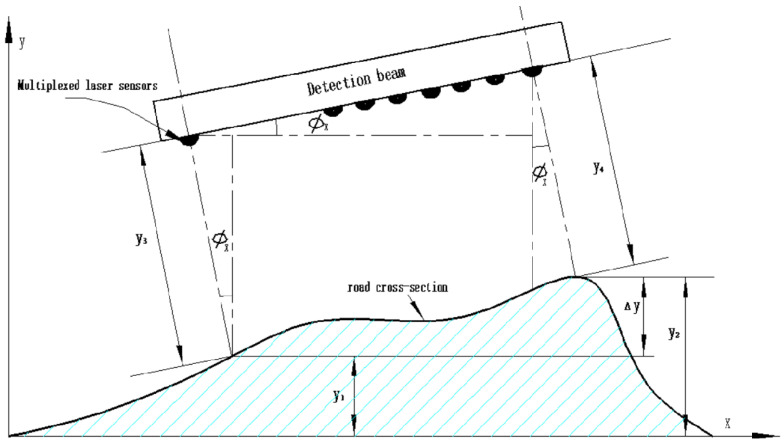
The Principle of Road Cross-section Data Acquisition.

**Figure 7 sensors-23-06758-f007:**
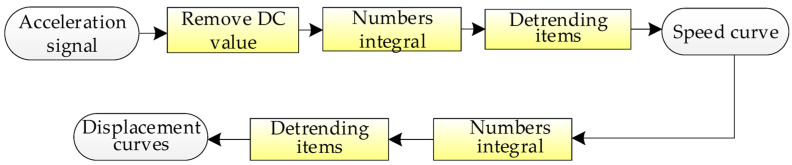
Acceleration Solving Velocity and Displacement Process.

**Figure 8 sensors-23-06758-f008:**
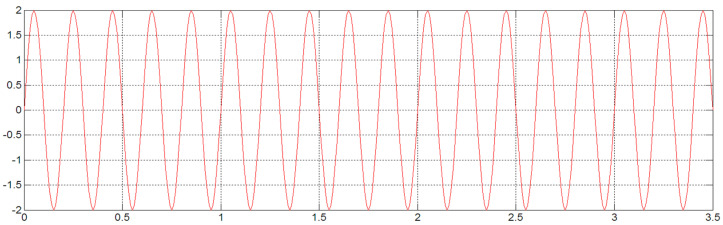
Original acceleration signal.

**Figure 9 sensors-23-06758-f009:**
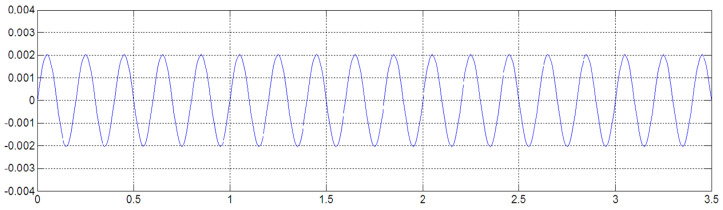
Displacement signal after Simpson quadratic integration.

**Figure 10 sensors-23-06758-f010:**
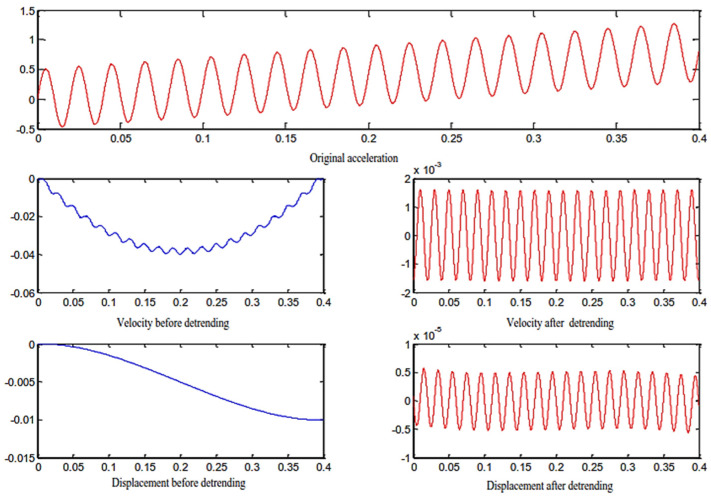
Before and after comparison of detrended items.

**Figure 11 sensors-23-06758-f011:**
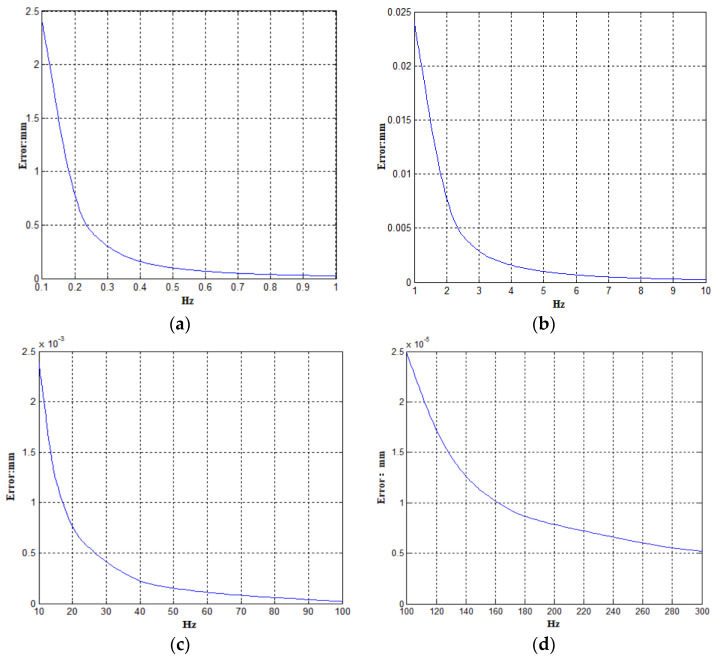
Measurement error curve of acceleration sensor. (**a**) 0.1~1 Hz error curve; (**b**) 1~10 Hz error curve; (**c**) 10~100 Hz error curve; (**d**) 100~300 Hz error curve.

**Figure 12 sensors-23-06758-f012:**
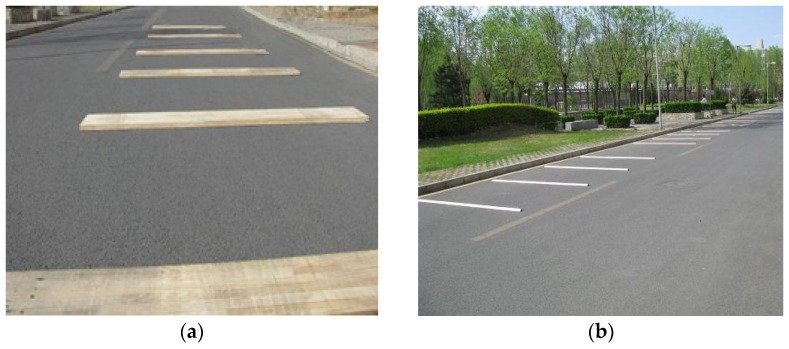
Pavement section block detection. (**a**) Planks; (**b**) Strips.

**Figure 13 sensors-23-06758-f013:**
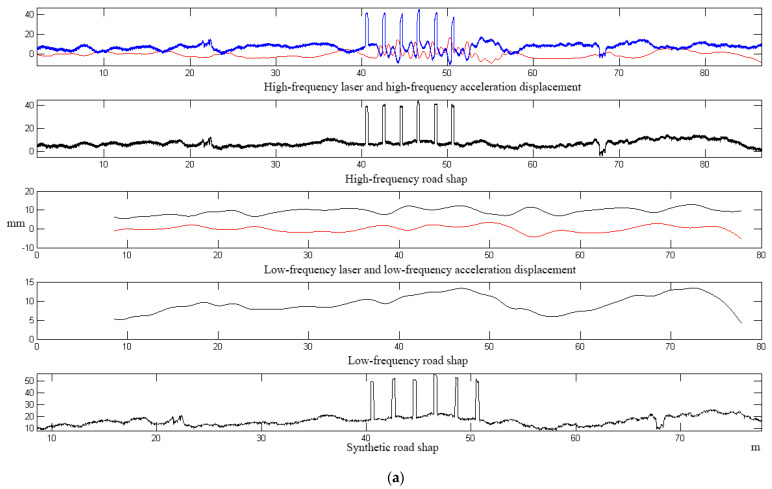
Curve of six planks at 30 km/h of the inspection vehicle. (**a**) The data curves of six planks; (**b**) Amplification curves of six planks; (**c**) Comparison curve of the 1st plank with the ideal shape.

**Figure 14 sensors-23-06758-f014:**
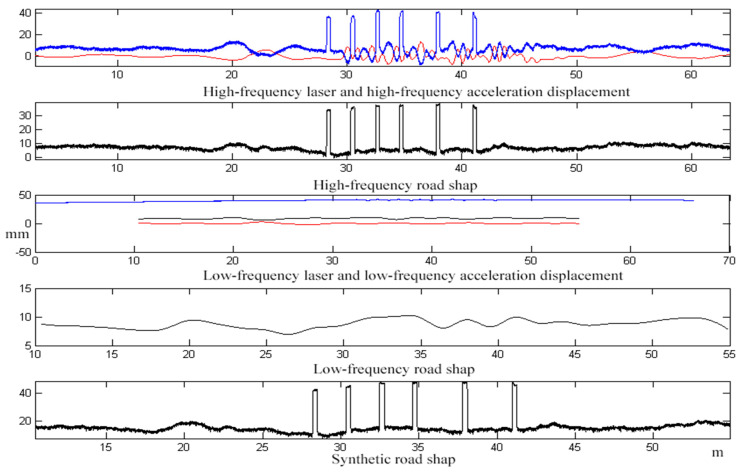
Curve of six planks at 60 km/h of the test vehicle.

**Figure 15 sensors-23-06758-f015:**
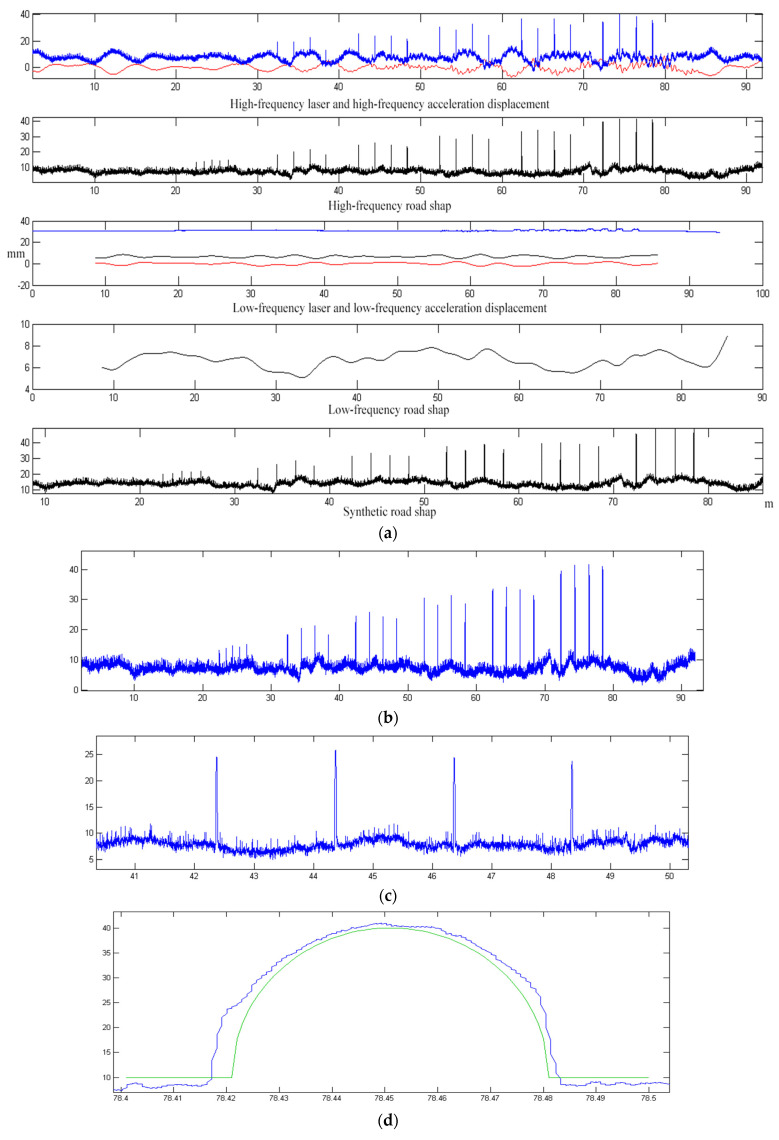
Curve of wooden strips at 30 km/h of the inspection vehicle. (**a**) The wooden strips data curve; (**b**) Curves of wooden strips; (**c**) Four 3 cm diameter semicircular wooden strips curves; (**d**) Comparison curves with the ideal shape at the 6 cm semicircular wooden strips.

**Table 1 sensors-23-06758-t001:** System comprehensive error relationship table.

Driving Speed (km/h)	Road Wavelength (m)	Maximum Measurement Error (mm)	Measuring Accuracy (%)	Road Frequency (Hz)
35	1	6.07	1.51%	9.7
	20	6.17	1.54%	0.48
	50	6.72	1.68%	0.19
	100	8.69	2.17%	0.22
50	1	6.07	1.51%	13.8
	20	6.12	1.53%	0.69
	50	6.19	1.83%	0.44
	100	7.35	1.84%	0.14
80	1	6.07	1.51%	22.2
	20	6.09	1.52%	1.1
	50	6.39	1.60%	0.28
	100	6.50	1.62%	0.1

**Table 2 sensors-23-06758-t002:** 30 km/h test results Table 3. 60 km/h test results.

Plank Position	Test Results (mm)			
	Transverse Length		Vertical Height	
1	3000	3002.5	30	34.8
2		3004.2		32.2
3		3001.7		30.9
4		2999.8		33.8
5		3005.3		29.7
6		3003.8		32.1
	Average	3002.8	Average	32.25
	Error	2.8	Error	2.25

**Table 3 sensors-23-06758-t003:** 60 km/h test results.

Plank Position	Test Results (mm)			
	Transverse Length		Vertical Height	
1	3000	3006.6	30	35.8
2		3006.5		33.2
3		3001.6		36.2
4		2998.3		29.8
5		3004.4		31.7
6		3005.6		34.1
	Average	3005.3	Average	33.5
	Error	5.3	Error	3.5

**Table 4 sensors-23-06758-t004:** Wooden strips test results.

**1 cm Semicircle**	**Test Results (mm)**			
**Strip Position**	**Transverse Length**		**Vertical Height**	
1	2000	2002.5	5	6.8
2		2004.2		6.2
3		2005.7		6.9
4		2000.8		5.8
5		2005.3		5.7
	Average	2003.7	Average	6.3
	Error	3.7	Error	1.3
**2 cm Semicircle**	**Test Results (mm)**			
**Strip Position**	**Transverse Length**		**Vertical Height**	
6	2000	2006.5	10	11.8
7		2006.2		12.2
8		2005.7		15.9
9		1998.8		9.8
	Average	2004.3	Average	12.3
	Error	4.3	Error	2.3
**3 cm Semicircle**	**Test Results (mm)**			
**Strip Position**	**Transverse Length**		**Vertical Height**	
10	2000	2005.5	15	14.8
11		2004.2		20.2
12		2006.7		19.9
13		2006.8		19.8
	Average	2005.8	Average	19.9
	Error	5.8	Error	4.9
**4 cm Semicircle**	**Test Results (mm)**			
**Strip Position**	**Transverse Length**		**Vertical Height**	
14	2000	2006.1	20	24.8
15		2006.9		26.2
16		2005.7		25.9
17		2001.8		24.3
	Average	2005.1	Average	25.3
	Error	5.1	Error	5.3
**5 cm Semicircle**	**Test Results (mm)**			
**Strip Position**	**Transverse Length**		**Vertical Height**	
18	2000	2007.1	25	30.8
19		2004.2		31.2
20		2006.7		30.9
21		2005.8		29.8
	Average	2005.9.	Average	30.7
	Error	5.9	Error	5.7
**6 cm Semicircle**	**Test Results (mm)**			
**Strip Position**	**Transverse Length**		**Vertical Height**	
22	2000	2006.5	30	36.8
23		2006.2		35.2
24		2005.7		36.9
25		2007.8		35.8
	Average	2006.5	Average	36.2
	Error	6.5	Error	6.2

## Data Availability

No need to explain.

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
