# Peer review of "Detection and Analysis of Pavement-Section Based on Laser Displacement Sensor"

_sensors, 2023, doi:10.3390/s23156758_

Round 1
Reviewer 1 Report
sensors-2482798:
Here are my comments:
The article is interesting; however, I request some additions.
*The English of the article must be proofread (by a native speaker).
*The introduction should be more scientific, contextualize the state of the art. In addition, clarify the paper's scientific contribution to the area of knowledge.
*In all data collected and presented on paper, present the error (standard deviation). After this, consider the measurement error in the discussion of the results.
*Confront with the existing literature (other measurement techniques) the results obtained in this research.
*Use statistics and reliability in interpreting the data.
*The conclusion is extensive; it should be shortened. The conclusion must concisely respond to the general objective of the research.
Moderate editing of English language required.
Author Response
Dear Reviewers:
Thank you for taking the time to review my manuscript over the holidays, and providing us with this great opportunity to submit a revised version of our manuscript. We appreciate the detailed and constructive comments provided by the reviewers. We have carefully revised the manuscript by incorporating all the suggestions by the review panel.
We hope this revised manuscript has addressed your concerns, and look forward to hearing from you.
Your Sincerely,
Jin Han
In response to your suggestion, I have made the following changes:
- The English of the article must be proofread (by a native speaker)
Response1: The author has corrected grammar, vocabulary and formatting throughout the article.
- The introduction should be more scientific, contextualize the state of the art. In addition, clarify the paper's scientific contribution to the area of knowledge.
Response2: The article adds a description of the current technology and its shortcomings in the introduction section, adding to the paper's contribution to this application area.
- In all data collected and presented on paper, present the error (standard deviation). After this, consider the measurement error in the discussion of the results.
Response3: The experimental tests in the fourth part of the article have been added, and the test data and errors are listed one by one according to the test conditions.
- Confront with the existing literature (other measurement techniques) the results obtained in this research.
Response4: The experimental results in the fourth part of the article analyze the applicability of the experiments in this paper.
- 5. Use statistics and reliability in interpreting the data.
Response5: Data graphs and data tables are added to the experimental cases in the fourth part of the article to support the authenticity and reliability of the data.
- The conclusion is extensive; it should be shortened. The conclusion must concisely respond to the general objective of the research.
Response6: The conclusion section has been optimized to highlight the innovations and application benefits of this paper..
Reviewer 2 Report
The authors proposed a Pavement-Section data acquisition and processing solution by combing multiple sensors including a laser displacement sensor, accelerometer, attitude sensor, and environmental monitors. A detailed discussion on the mechanism behind this solution has been provided and the analysis of the error is investigated thoroughly. This method is expected to be efficient in detecting the road condition in airports and highways. The obtained results and topic are interesting for the broad readership of Sensors. However, there are some crucial information and discussions missing in the manuscript. Before this manuscript becomes acceptable, the authors are required to address the following comments well.
1, The English in this manuscript is not good. There are lots of confusing paragraphs and sentences. The quality of the writing should be significantly improved. Some examples can be found in line 13, 24, 26, 28, 122-127, 215-216, 233, 322, and so on.
2, There are lots of typos in the manuscript. The authors need to revise the manuscript carefully. Some examples are in line 40, 536, 543, and so on.
3, The authors are recommended to add a SPACE between value and unit.
4, KHz should be kHz.
5, The novelty of this work should be strengthened. What are the innovative points or improvements compared to conventional solutions? Sensor? Algorithm? Or others?
6, The mark in Figure 4 is misleading. If following those marks, h4cos(theta)-h2 should be 0 in Equation (1). Please clarify this concern.
7, What is the definition of a(t) in Equation (2)?
8, There is a misleading cross mark in Figure 6.
Based on these comments, this manuscript is recommended for major revision. A revised manuscript is required.
Please check the comments.
Author Response
Dear Reviewers:
Thank you for taking the time to review my manuscript over the holidays, and providing us with this great opportunity to submit a revised version of our manuscript. We appreciate the detailed and constructive comments provided by the reviewers. We have carefully revised the manuscript by incorporating all the suggestions by the review panel.
We hope this revised manuscript has addressed your concerns, and look forward to hearing from you.
Your Sincerely,
Jin Han
In response to your suggestion, I have made the following changes:
- The English in this manuscript is not good. There are lots of confusing paragraphs and sentences. The quality of the writing should be significantly improved. Some examples can be found in line 13, 24, 26, 28, 122-127, 215-216, 233, 322, and so on
Response1: The author has corrected grammar, vocabulary and formatting throughout the article.
- There are lots of typos in the manuscript. The authors need to revise the manuscript carefully. Some examples are in line 40, 536, 543, and so on..
Response2: Typos in the manuscript have been corrected and the corrections have been marked.
- The authors are recommended to add a SPACE between value and unit..
Response3: Add spaces as required.
- KHz should be kHz.
Response4: T Modified as suggested.
- 5. The novelty of this work should be strengthened. What are the innovative points or improvements compared to conventional solutions? Sensor? Algorithm? Or others?
Response5: The innovations and application advantages of this paper are summarized in the introduction and conclusion sections.
- The mark in Figure 4 is misleading. If following those marks, h4cos(theta)-h2 should be 0 in Equation (1). Please clarify this concern.
Response6: The wrong dimension in Fig. 4 was modified and it matches Eq. (1) correctly.
- What is the definition of a(t) in Equation (2)?
Response7: The definition of a(t) has been added to the paper.
- 8. There is a misleading cross mark in Figure 6.
Response8: Misleading signs have been removed.
Reviewer 3 Report
The paper is not very clear and handles an issue that I feel is well covered already in the literature. The simple detection of the surface and cross section is not something requiring huge advances in research and the paper does not indeed examine the extensive past research on the topic and the field on the whole. For it to be a scientific paper needs to focus on using the technology for something more novel than simple road detection. Overall the paper feels lacking in scientific merit. A more extended view on the literature would show this and maybe point the authors in a better direction.
Moderate changes to english required
Author Response
Dear Reviewers:
Thank you for taking time out of your vacation to review my manuscript and for the excellent opportunity to submit a revised version of the manuscript. From the comments you have given, it is clear that you are well researched in the field of road testing and are an expert in the relevant field, and it is an honor to receive your advice. Thank you for your detailed and constructive comments. We have carefully read your suggestions, absorbed all the suggestions from the reviewers, and revised the manuscript carefully.
We hope this revised manuscript has addressed your concerns, and look forward to hearing from you.
Your Sincerely,
Jin Han
In response to your suggestion, I have made the following changes:
The paper is not very clear and handles an issue that I feel is well covered already in the literature. The simple detection of the surface and cross section is not something requiring huge advances in research and the paper does not indeed examine the extensive past research on the topic and the field on the whole. For it to be a scientific paper needs to focus on using the technology for something more novel than simple road detection. Overall the paper feels lacking in scientific merit. A more extended view on the literature would show this and maybe point the authors in a better direction.
Response: Although the research in this paper has been elaborated in the relevant literature, most of them are limited to the detection of single data from a single sensor, with low detection accuracy, generally at the centimeter level. Some of them, although using the vehicle-mounted method, have limited detection range (only one lane can be detected at a time) and the detection accuracy cannot be guaranteed at high speeds (vehicle speed >50km/h). In this paper, due to the use of multi-sensor fusion method, and the acceleration signal filtering process, can meet the accuracy of high-speed detection. In addition, the detection vehicle in this paper can adjust the width of the detection beam according to the width of the road surface, and can detect two lanes at the same time. Therefore, the methods and techniques designed in this paper are innovative and practical.
Round 2
Reviewer 1 Report
The authors performed a good review on the paper.
Moderate editing of English language required.
Author Response
Dear Reviewer:
Thank you for taking time out of your busy schedule to review my manuscript and for the great opportunity to submit a revised version of the manuscript. We are very grateful to the reviewers for their detailed and constructive comments. We have incorporated your requests for English editing and have carefully revised the manuscript. We hope that this revised manuscript will address your concerns and look forward to hearing from you.
Best regards,
Yours sincerely,
Jin Han
Reviewer 2 Report
The authors have answered part of the previous comments. There are some concerns left in the manuscript. Before this manuscript becomes acceptable, the authors are required to address the following comments well.
1, For all the figure captions, if the authors are willing to capitalize the first letter for every word, please make it consistent for all figures.
2, Please go through the whole manuscript and make sure to add a SPACE between the value and unit.
3, The definition of a(t) is still missing when it is first cited in the manuscript.
4, Configuration and alignment of Equation 6 and 10 should be improved.
5, From line 305 to line 306, please revise the sentence to improve the quality of English.
6, What’s the correlation between Equation 13 and Figure 10? Does the original acceleration represent Equation 13? Why do you have two types of velocity before detrending? In addition, please clarify whether such a detrending operation introduces interruptions in the magnitude of acceleration or not.
7, In line 339, it should be “Equation (14)”.
8, In Figure 11, some values cannot match Equation 14. For example, in (a), when f is 0.4 Hz, the error should be 0.155 rather than 2.5 based on Equation 14. Please check the scale of the axis.
9, In line 390, the calculation result is incorrect. Please modify that.
10, There are some measuring accuracy values in Table 1 inconsistent with Equation 20. For example, when the driving speed is 35 km/h, and the road wavelength is 100 m, the measuring accuracy based on Equation 20 should be 2.3% rather than 1.64%. Please clarify these discrepancies.
11, For Figure 8, 9, 11, 13, and 15, please re-plot them with a consistent format and do not just use the screen captures of MATLAB. All figures in this manuscript should have a consistent font format as well.
12, The image resolution of Fig. 13(a), Fig. 14, and Fig. 15(a) is poor. Please re-plot them to improve the resolution.
13, In Fig. 15(a), please remove the coordinates mark.
Based on the abovementioned comments, this manuscript is recommended for major revision. A revised version is required.
Author Response
Dear Reviewers:
Thank you for taking time out of your busy schedule to review my manuscript and for the great opportunity to submit a revised version of the manuscript. We are very grateful to the reviewers for their detailed and constructive comments. We have carefully revised the manuscript by incorporating all the suggestions by the review panel.
We hope this revised manuscript has addressed your concerns, and look forward to hearing from you.
Best regards,
Your Sincerely,
Jin Han
In response to your suggestion, I have made the following changes:
1.For all the figure captions, if the authors are willing to capitalize the first letter for every word, please make it consistent for all figures.
Response 1: The formatting of all graphic titles has been corrected in a uniform manner.
2.Please go through the whole manuscript and make sure to add a SPACE between the value and unit.
Response 2: Added the SPACE between the value and unit.
3.The definition of a(t) is still missing when it is first cited in the manuscript.
Response 3: Definition of a(t) added at line 151.
4.Configuration and alignment of Equation 6 and 10 should be improved.
Response 4: Formatting and placement corrections were made to Equations 6 and 10.
5.From line 305 to line 306, please revise the sentence to improve the quality of English.
Response 5: Sentences 305-306 were revised.
6.What’s the correlation between Equation 13 and Figure 10? Does the original acceleration represent Equation 13? Why do you have two types of velocity before detrending? In addition, please clarify whether such a detrending operation introduces interruptions in the magnitude of acceleration or not.
Response 6: Equation 13 and Figure 10 are related; this paper uses Equation 13 only as an example of how to remove the trend term rather than the actual acceleration at the time of detection; in this paper, the removal of the trend term by setting the polynomial fit order n is continuous and does not result in a break in acceleration amplitude.
7.In line 339, it should be “Equation (14)”.
Response 7: Modify to “Equation 14”
8.In Figure 11, some values cannot match Equation 14. For example, in (a), when f is 0.4 Hz, the error should be 0.155 rather than 2.5 based on Equation 14. Please check the scale of the axis.
Response 8: The numerical axis scale of Fig. 11(a) was modified to match the values to Eq. 14.
9.In line 390, the calculation result is incorrect. Please modify that.
Response 9: Corrected the miscalculated value of line 390.
10.There are some measuring accuracy values in Table 1 inconsistent with Equation 20. For example, when the driving speed is 35 km/h, and the road wavelength is 100 m, the measuring accuracy based on Equation 20 should be 2.3% rather than 1.64%. Please clarify these discrepancies.
Response 10: Corrections made to the miscalculated values in Table I to match Eq. 20.
11.For Figure 8, 9, 11, 13, and 15, please re-plot them with a consistent format and do not just use the screen captures of MATLAB. All figures in this manuscript should have a consistent font format as well.
Response 11: The graphical format of this paper was modified to make it uniform.
12.The image resolution of Fig. 13(a), Fig. 14, and Fig. 15(a) is poor. Please re-plot them to improve the resolution.
Response12: An upgrade was made to the listed graphics resolution.
13.In Fig. 15(a), please remove the coordinates mark.
Response13: The Fig. 15(a) coordinates mark. have been removed.
Reviewer 3 Report
I appreciate the notes from the authors on the revisions. I still however find the paper to be lacking in its total merit and practicality. The focus on being able to detect 2 lanes seems to be the most relevant factor in the paper being innovative. If the paper moves forward I'd say maybe more focus on this aspect can be added and comparisons to previous work where the limitation exists can be more clearly added to the paper.
English is fine
Author Response
Dear Reviewer:
Thank you for taking time out of your busy schedule to review my manuscript and for the great opportunity to submit a revised version. The reviewers' detailed and constructive comments are greatly appreciated. We have incorporated your suggestions, corrected and clarified the innovations in this paper, and carefully revised the manuscript.
We hope that this revised manuscript will address your concerns and look forward to hearing from you.
With best regards,
Yours sincerely
Han Jin
In response to your suggestion, I have made the following changes:
lacking in its total merit and practicality. The focus on being able to detect 2 lanes seems to be the most relevant factor in the paper being innovative. If the paper moves forward I'd say maybe more focus on this aspect can be added and comparisons to previous work where the limitation exists can be more clearly added to the paper.
Response: The innovations are described in the introduction and conclusion sections.